# Sulfonamidoboronic Acids as “Cross-Class” Inhibitors of an Expanded-Spectrum Class C Cephalosporinase, ADC-33, and a Class D Carbapenemase, OXA-24/40: Strategic Compound Design to Combat Resistance in *Acinetobacter baumannii*

**DOI:** 10.3390/antibiotics12040644

**Published:** 2023-03-24

**Authors:** Maria Luisa Introvigne, Trevor J. Beardsley, Micah C. Fernando, David A. Leonard, Bradley J. Wallar, Susan D. Rudin, Magdalena A. Taracila, Philip N. Rather, Jennifer M. Colquhoun, Shaina Song, Francesco Fini, Kristine M. Hujer, Andrea M. Hujer, Fabio Prati, Rachel A. Powers, Robert A. Bonomo, Emilia Caselli

**Affiliations:** 1Department of Life Sciences, Università di Modena e Reggio Emilia, Via Campi 103, 41125 Modena, Italy; 2Department of Chemistry, Grand Valley State University, Allendale, MI 49401, USA; 3Department of Medicine, Case Western Reserve University School of Medicine, Cleveland, OH 44106, USA; 4Research Service, Louis Stokes Cleveland Department of Veterans Affairs Medical Center, Cleveland, OH 44106, USA; 5Research Service, Atlanta Veterans Medical Center, Decatur, GA 30033, USA; 6Department of Microbiology and Immunology, Emory University School of Medicine, Atlanta, GA 30307, USA; 7Emory Antibiotic Resistance Center, Emory University School of Medicine, Atlanta, GA 30307, USA; 8Louis Stokes Cleveland Department of Veterans Affairs Medical Center, Cleveland, OH 44106, USA; 9Departments of Medicine, Pharmacology, Molecular Biology and Microbiology, Biochemistry, Proteomics and Bioinformatics, Case Western Reserve University School of Medicine, Cleveland, OH 44106, USA; 10CWRU-Cleveland VAMC Center for Antimicrobial Resistance and Epidemiology (Case VA CARES), Cleveland, OH 44106, USA

**Keywords:** antibiotic resistance, *β*-lactamases, *Acinetobacter baumannii*, boronic acids, cross-class inhibitors

## Abstract

*Acinetobacter baumannii* is a Gram-negative organism listed as an urgent threat pathogen by the World Health Organization (WHO). Carbapenem-resistant *A. baumannii* (CRAB), especially, present therapeutic challenges due to complex mechanisms of resistance to *β*-lactams. One of the most important mechanisms is the production of *β*-lactamase enzymes capable of hydrolyzing *β*-lactam antibiotics. Co-expression of multiple classes of *β*-lactamases is present in CRAB; therefore, the design and synthesis of “cross-class” inhibitors is an important strategy to preserve the efficacy of currently available antibiotics. To identify new, nonclassical *β*-lactamase inhibitors, we previously identified a sulfonamidomethaneboronic acid **CR167** active against *Acinetobacter*-derived class C *β*-lactamases (ADC-7). The compound demonstrated affinity for ADC-7 with a *K*_i_ = 160 nM and proved to be able to decrease MIC values of ceftazidime and cefotaxime in different bacterial strains. Herein, we describe the activity of **CR167** against other *β*-lactamases in *A. baumannii*: the cefepime-hydrolysing class C extended-spectrum *β*-lactamase (ESAC) ADC-33 and the carbapenem-hydrolyzing OXA-24/40 (class D). These investigations demonstrate **CR167** as a valuable cross-class (C and D) inhibitor, and the paper describes our attempts to further improve its activity. Five chiral analogues of **CR167** were rationally designed and synthesized. The structures of OXA-24/40 and ADC-33 in complex with **CR167** and select chiral analogues were obtained. The structure activity relationships (SARs) are highlighted, offering insights into the main determinants for cross-class C/D inhibitors and impetus for novel drug design.

## 1. Introduction

The global increase in antibiotic resistance has led to a critical need for antibacterial drug research and development efforts. According to the U.S. Centers for Disease Control and Prevention (CDC), if new antibiotics are not developed by 2050, the number of annual deaths from multi-drug-resistant bacteria will reach 10 million, more than for cancer and heart disease combined [1]. Even if prevention measures are able to reduce the incidence of certain infections, the impact of COVID-19 has contributed to the spread of healthcare-associated and antibiotic-resistant infections. The Centers for Diseases Control and Prevention (CDC) recently issued a special report on the US impact of COVID-19 on antimicrobial resistance that identified an overall 35% increase in carbapenem-resistant infections and an alarming 78% increased rate for nosocomial acquired infections [2]. The World Health Organization (WHO) has also drawn up a priority list of pathogens and assigned the highest priority to the Gram-negative bacteria *Acinetobacter baumannii*, *Pseudomonas aeruginosa,* and *Enterobacterales* species resistant to carbapenems, defining them as an alarming threat to public health [3]. *A. baumannii* is responsible for many infections; e.g., bloodstream infections, urinary tract infections, respiratory infections, skin and soft tissue infections, and meningitis [4,5]. This pathogen is often resistant to all classes of antibiotics. *A. baumannii* possesses multiple antibiotic resistance determinants, such as *β*-lactamase enzymes, efflux pumps, and modifications in the *β*-lactam target PBPs, as well as a modified outer membrane that decreases permeability to antimicrobials. Moreover, this pathogen is also able to acquire other determinants through genetic exchange, natural transformation, and outer membrane vesicles [6].

*β*-Lactamases are enzymes able to hydrolyze *β*-lactams, thereby impairing their inhibitory activity against penicillin-binding proteins (PBPs). They are divided into four classes: A, B, C, and D. Class A, C, and D enzymes are serine *β*-lactamases, while class B are metallo-enzymes because they possess one or two zinc ions in their catalytic sites [7]. The prevalence of *β*-lactamases, and their ability to constantly evolve to hydrolyze new classes of *β*-lactams, makes *β*-lactamase inhibition an important strategy to preserve the efficacy of currently available antibiotics. To face often pan-drug-resistant organisms, such as *A. baumannii*, a strategy that considers the most common *β*-lactamases occurring in the pathogen, rather than focusing on a single class of enzymes [8], is warranted. Currently, the overall number of *β*-lactamase enzymes identified is 7739 (according to the *β*-lactamase database BLDB; last update: 1 November 2022) [9], and among them, class C cephalosporinases and class D carbapenemases are of particular concern for *A. baumannii* infections. The chromosomally encoded *Acinetobacter*-derived cephalosporinases (ADCs) are intrinsic to *A. baumannii* strains, and they confer resistance to cephamycins (cefoxitin and cefotetan), cephalosporins, penicillins, and monobactams. Moreover, ADCs are resistant to common *β*-lactamase inhibitors, such as clavulanate and sulbactam [8,10]. Unfortunately, avibactam, a new diazabicyclooctane (DBO) inhibitor, is also unable to restore susceptibility. The class C *β*-lactamase ADC-7 was one of the first ADCs identified from an *A. baumannii* clinical isolate and was shown to possess a high turnover rate for narrow-spectrum cephalosporins and low affinity for clavulanic acid, sulbactam, and tazobactam inhibitors [11]. More recently, a survey of carbapenem-resistant *Acinetobacter* isolates revealed the prominence of ADC-33, which can hydrolyze expanded-spectrum cephalosporins (e.g., ceftazidime (CAZ) and cefepime (FEP)) and aztreonam but not carbapenems [8]. These *β*-lactamases are regarded as expanded-spectrum AmpCs (ESACs).

The active sites of class C *β*-lactamases accommodate the bulky extended-spectrum cephalosporins. The class C active sites can be divided into R1 and R2 subsites. R1 is the area where the R1 side chain at C7 (C6) of the *β*-lactam core (Figure 1) interacts, while the R2 site accommodates the R2 C3 (C2) chain of *β*-lactams in the opposite region of the active site [12]. The X-ray crystal structure of the acyl complex of ADC-7 and ceftazidime reveals how the amide group at C7 interacts with conserved residues Gln120 and Asn152, and the carboxylate present at C3 (C4) is recognized by a polar region comprised of Ser318, Arg343, and Asn346 (throughout the manuscript, SANC numbering is used for all ADC amino acids) [13].

Class D *β*-lactamases (oxacillin-hydrolyzing (OXA) enzymes) are widespread in *A. baumannii* and confer resistance not only to penicillins and cephalosporins but also against carbapenems which are considered the last-resort *β*-lactam antibiotic therapy. Of particular concern are OXA-23 and OXA-24/40 *β*-lactamases, which are carbapenem-resistance determinants, thus limiting treatment options. Often, synergistic use of antibiotics (combination therapy) is needed to overcome resistance, which makes single therapies (monotherapy) ineffective [14]. The most important features differentiating class D *β*-lactamases from other serine *β*-lactamases are their hydrophobic active sites and, in OXA-23 and OXA-24/40, a peculiar “tunnel-like” entrance to the catalytic pocket. This structure was first identified in OXA-24/40, where it is formed through the specific arrangement of Tyr112 and Met223 side chains. These residues create a barrier that deters the entrance of antibiotics with bulky substituents, while smaller substituents, such as the hydroxyethyl side chains of carbapenems, are readily accommodated [15]. 

In general, *β*-lactam-based *β*-lactamase inhibitors, such as clavulanic acid, do not inhibit OXA enzymes and, therefore, do not have efficacy against resistant *A. baumannii* strains. Interestingly, a novel *β*-lactam derivative, LN-1-255, did inhibit carbapenem-hydrolyzing class D *β*-lactamases, restoring carbapenem activity against critical *A. baumannii* strains (Figure 2) [16,17]. Non-*β*-lactam *β*-lactamase inhibitors, such as avibactam (see above), relebactam, and vaborbactam, are not effective against class D *β*-lactamases [14]; however, the more recent diazabicyclooctane, durlobactam, exhibited improved activity [18,19].

Boronic acid transition state inhibitors (BATSIs) have recently been developed as inhibitors of *β*-lactamases. The nucleophilic attack of the catalytic serine on the boron atom forms a tetrahedral adduct, which mimics the acylation (or deacylation) tetrahedral structure of the high-energy intermediate formed during *β*-lactam hydrolysis. Boronic acids are known to inhibit class A and class C *β*-lactamases with *K*_i_ values in the nM range [20,21], while relatively few compounds demonstrate activity against class D enzymes [22,23,24]. 

The first BATSI to reach the market, vaborbactam, is a mono-cyclic boronic acid (Figure 2) that has proved to be an excellent class A and C *β*-lactamase inhibitor, although it is not active against class D enzymes [19]. However, the bicyclic boronate core structure exhibits cross-class inhibition, with two such inhibitors, QPX7728 and taniborbactam, currently in phase 1 and phase 3 clinical trials, respectively [25,26]. Other studies demonstrated the ability of select boronic acids to inhibit OXA enzymes [22,27,28], but none were further developed. In 2010, the work performed by Tan et al. revealed the activity of arylsulfonamide boronic acids to be in the low µM range against five different *β*-lactamases [22]. In particular, the 4,7-dichloro-1-benzothien-2-yl-sulfonylaminomethyl boronic acid (DSABA, Figure 2) demonstrated improved affinity for OXA-24/40 (IC_50_ = 5.6 µM), while also inhibiting class A and C *β*-lactamases with sub- to low μM IC_50_ values. Other sulfonamide boronic acids have proved to be excellent inhibitors of the class C *β*-lactamase AmpC from *E. coli*, displaying a distinct structure–activity relationship when compared with the better characterized carboxamides. Amongst the different benzyl sulfonamide derivatives, **CR167** (Figure 2)**,** bearing a carboxylate in the *meta* position of the aromatic ring, has emerged as one of the most potent AmpC inhibitors (*K*_i_ = 1.3 nM) [29,30]. The compound was also tested against class A enzymes and proved to have potent activity against SHV-1 (IC_50_ = 0.43 µM) and KPC-2 (IC_50_ = 0.20 µM) *β*-lactamases [31].

Encouraged by the results obtained against class A and C *β*-lactamases, **CR167** was tested against an important ADC variant found in *A. baumannii*. The compound demonstrated enhanced affinity for ADC-7 with a *K*_i_ = 160 nM and proved to be able to decrease MIC values of CAZ (reducing the MIC for CAZ in *E. coli* DH10B pBCSK, *bla*_ADC-7_ from 64 mg/L to 1 mg/L) [32]. The X-ray structure of the complex ADC-7/**CR167** (PDB 5WAE) was obtained and the mechanism by which this inhibitor extends into the R1 binding site, where the side chain at C7 of the cephalosporin interacts, was demonstrated. Indeed, the sulfonamide formed hydrogen bonds with Asn152 and Gln120, mimicking those formed with the amide group at C7/C6 of the *β*-lactam. The benzyl ring stacks with Tyr221 and, in this way, the carboxylate is positioned in a region composed of the polar residues Asn212, Ser320, and Thr319 on the opposite side of the catalytic pocket with respect to the canonical binding region of the carboxylate in C3 (C4) of *β*-lactams, which is composed of Ser318, Arg343, and Asn346. 

In this work, we describe the activity of **CR167** against critical *β*-lactamases in *A. baumannii* ADC-33 (class C) and OXA-24/40 (class D), as well as our attempts to further improve its activity, and show that it is a valuable cross-class (C and D) inhibitor. The structure of the complex of OXA-24/40 with **CR167** was obtained and compared with that of the carbapenem doripenem, suggesting the rational design of five chiral analogues that were synthesized. The synthesis, kinetics, crystallographic structures, and microbiological activity of select compounds are reported, offering insights into the main determinants for cross-class C/D inhibitors. 

## 2. Results

### 2.1. CR167 as an Inhibitor of Class C and Class D β-Lactamases

To characterize a cross-class inhibitor for the class C and D *β*-lactamases of *A. baumannii*, we assessed the activity of the achiral BATSI **CR167** against two major resistance determinants. **CR167** was synthesized as previously described [30] and tested against the class C enzyme ADC-33 and the class D enzyme OXA-24/40.

Kinetic assays

The affinity of **CR167** for the two different *β*-lactamases was measured in competition assays using nitrocefin (NCF) as the indicator substrate (Table 1). Competitive inhibition in the nM range was observed against ADC-33 (0.057 μM) and in the low μM range (4.4 μM) against OXA-24/40, confirming this compound as a potent inhibitor of class C enzymes and suggesting that the sulfonamidoboronate might be a potential scaffold for cross-class inhibition with the class D enzymes. 

Crystallographic structures

The X-ray crystal structure of OXA-24/40 in complex with compound **CR167** was determined to 2.01 Å resolution (Table 2) and compared with the structure of OXA-24/40 bound in an acyl-enzyme complex with the carbapenem doripenem (7RPF) [33], as well as with the previously reported ADC-7 (class C) complex with **CR167** (5WAE) [32].

In its complex with OXA-24/40, **CR167** adopts a binding mode that is similar to that of the carbapenem doripenem and extends away from the intact Tyr112/Met223 bridge into a hydrophobic pocket composed of Met114, Tyr112, Met223, and Trp221, referred to as the R2 site (Figure 3 and Figure 4). The aromatic ring of the inhibitor is positioned such that it interacts with Tyr112 in an edge-to-face stacking interaction. The centroid–centroid distance between the aryl rings of the inhibitor and Tyr112 is 5.6 Å, with an angle of 89° between the planes of the aryl rings. As expected, the boron atom is covalently attached to the Oγ of the catalytic Ser81 (Figure 4). The O1 hydroxyl of the boronic acid is bound in the presumed oxyanion hole, whereas the O2 hydroxyl interacts with Ser128 and a water molecule present in the active site. As in the complex with doripenem, the terminal carboxylate group of **CR167** forms a key ionic interaction with the class D carboxylate recognition residue Arg261 and a hydrogen bond with Ser219. From this superposition, we observed that the R2 site is relatively open and offers space for the design and optimization of inhibitors, whereas the region surrounding the C(6)-hydroxyethyl group of carbapenems does not offer much space to expand.

Next, the structures of the complexes of **CR167** with OXA-24/40 and ADC-7 (PDB 5WAE) were superposed (Figure 5)**.** Interestingly, the binding mode of the inhibitor in the two different enzyme classes was opposite, with **CR167** extending into the R1 region of ADC-7 where the C7 amide group of *β*-lactams is accommodated. Indeed, in ADC-7, the sulfonamide group mimics the R1 amide group, forming interactions with the same residues (Asn152 and Gln120), and the carboxylate interacts with Asn212 and Ser320 on the opposite side of the active site from where the C4 carboxylate of β-lactams binds. In the OXA-24/40 complex, **CR167** binds in the R2 region of the active site, forming an essential interaction with its carboxylate recognition residue Arg261, and resembles the carbapenem binding orientation observed in class D enzymes [34]. Despite the overall binding mode difference between the *β*-lactamase classes, superposition of these three complexes revealed that the carbon α to the boron of **CR167** is in the same position as the C(6) atom of doripenem where the hydroxyethyl side chain is attached. The hydroxyethyl group can be recognized from a hydrophobic pocket in OXA24/40 formed by Val130, Trp167, and Leu168 (Figure 3B). Notably, this group is essential to the carbapenem class of antibiotics, which bind with high affinity to the class D enzymes [35]. Therefore, this position offered a possible site to improve affinity for the class D enzymes by modifying **CR167** to better mimic carbapenems through the addition of a substituent at this position. However, considering that space is limited in this region—particularly for OXA-24/40—we hypothesized that a small alkyl chain (methyl or ethyl) could be inserted in this position, making this carbon atom stereogenic, and predicted that the better fit would be for the (S) enantiomer.

### 2.2. Chiral Analogues of CR167 

**CR167** was shown to inhibit both *Acinetobacter* class C and D *β*-lactamases. Based on our structural data, we designed a series of five chiral α-sulfonamidoboronic acid analogues of **CR167** to improve activity as “cross-class” inhibitors (Table 1). The benzylsulfonamide group was conserved because this group interacted with important residues—especially in class C enzymes, where it is in the same position as the R1 amide group of *β*-lactams—and analogues with and without the *meta*-carboxylate were obtained. To improve inhibition towards OXA-24/40, we inserted a methyl or ethyl group as a substituent on the carbon attached to the boron, aiming to mimic the hydroxyethyl group of carbapenems (indeed, *β*-hydroxylboronates are not stable) [36]. The configuration at the C6 position of carbapenems is opposite to that of the corresponding C7 position in cephalosporins (Figure 1). Therefore, all chiral analogues were synthesized in the (*S*) absolute configuration, except for compound **6e**, which was also obtained in the opposite (*R*) configuration to assess stereochemical preferences between class C and D *β*-lactamases. 

Synthesis

The synthetic pathways for α sulfonamidoboronic acids **6a–e** are summarized in Figure 1. Commercially available alkylboronic acids were converted in corresponding pinanendiol esters **1a–c**; (+)- or (−)- pinanendiol was used as a chiral auxiliary [29,36]. The stereoselective homologations of compounds **1a–c** to the corresponding α-chloroalkylboronates **2a–c** (90–97% yield) were carried out at −100 °C in anhydrous THF through a reaction with dichloromethylithium formed in situ from anhydrous CH_2_Cl_2_ and *n*-BuLi. Crude chloroalkylboronates were allowed to react at −100 °C with lithium bis(trimethylsilyl)amide in dry THF to afford compounds **3a–c** (56–85% yield from **1**) [29]. The hydrolysis to the corresponding hydrochloride salts **4a-–c** was accomplished using HCl in dioxane (91–100% yield). Two different commercially available sulfonyl chlorides (benzylsulfonyl chloride and *m*-methoxycarbonylbenzylsulfonyl chloride) were used to obtain α-sulfonamidoboronates **5a–e** (52–76% yield). Simultaneous deprotection of both carboxylic and boronic groups in **5b**, **5d,** and **5e** was achieved through acidic hydrolysis, affording **6b** (85% yield), **6d** (42%), and **6e** (52%), while deprotection of boronic groups in **5a** and **5c** was accomplished through transesterification with isobutylboronic acid, affording **6a** (75%) and **6c** (66%).

Inhibition kinetics

The binding affinity of each of the compounds **6a–e** with ADC-33 and with OXA-24/40 was determined after 5 min pre-incubation of boronic acid and the enzyme, using competition assays with nitrocefin (NCF) as the substrate. The *K*_i_ values for all BATSIs, corrected for NCF affinity, are reported in Table 1. For both enzymes, the best inhibitor of the group was **6b** (*K*_i_ = 0.79 and 5.2 μM for ADC-33 and OXA-24/40, respectively). The worst inhibitor for both was **6c** (*K*_i_ = 193 and > 1000 μM). Compounds **6a** and **6e** demonstrated lower affinity for OXA-24/40 (*K*_i_ = 630 and > 1000 μM, respectively), but they were among the best inhibitors against ADC-33 (*K*_i_ = 3.1 and 3.7 μM, respectively). These compounds differ in three structural characteristics: stereochemistry, the presence of the *m*-carboxylate on the aromatic ring, and the small chain linked to the chiral center. When compounds (*S*)-**6d** and (*R*)-**6e,** which differ only in the configuration of the stereogenic center, were compared in ADC-33, (*R*)-**6e** (*K*_i_ = 3.7 μM) was found to be more potent than (*S*)-**6d** (*K*_i_ = 22 μM). Conversely, in OXA-24/40, **6d** (*K*_i_ = 320 μM) has a higher affinity than **6e** (*K*_i_ > 1000 μM). The *m*-carboxylate is important for inhibition of both enzymes, but it is more critical for the inhibition of OXA-24/40 than for ADC-33; indeed, the affinity of compound **6b**, which contains the *m*-carboxylate (*K*_i_ = 5.2 μM), for OXA-24/40 is 120-fold higher than the affinity of compound **6a,** which lacks this group (*K*_i_ = 630 μM), while the difference for ADC-33 is only 4- to 8-fold. Finally, the smaller the steric hindrance is at the R1 position, the higher the affinity is. Furthermore, in this case, the nature of the R1 substituent had a greater impact for the inhibition of OXA-24/40 as compared to ADC-33. In fact, **6b** (R1 = methyl) is 60-fold more potent than **6d** (R1 = ethyl) in the case of OXA-24/40 (5.2 μM for **6b** and 320 μM for **6d**) but only 28-fold in the case of ADC-33 (0.79 μM for **6b** and 22 μM for **6d**).

Structural analysis

The X-ray crystal structures of ADC-33 and OXA-24/40 in complex with compounds (*S*)-**6d** and (*R*)-**6e** were determined to resolutions between 1.47 and 1.55 Å (Table 2).

ADC-33 crystallized as a dimer and OXA-24/40 as a monomer in the asymmetric unit. In each structure, initial *F*_o_–*F*_c_ difference electron-density maps (contoured at 3σ) indicated the presence of the inhibitors in the active sites, and continuous density with the Oγ of the catalytic serine residues suggested they are covalently bound, as expected. Polder omit maps confirmed the conformations of the inhibitors in the final models (Appendix A). These inhibitors were selected because they were the enantiomers that showed differences in binding affinity between the two β-lactamase classes. For ADC-33, the stereochemical arrangement in **6e** is preferred, and in OXA-24/40, **6d** is preferred. We compared these structures with the ADC-7 and OXA-24/40 complexes with **CR167**. In all three ADC complexes (Figure 6), the inhibitor adopts a relatively similar conformation, extending into the R1 binding site. The sulfonamide forms hydrogen bonds with Asn152 and Gln120, mimicking those formed with the C7 amide group of the cephalosporins. Additionally, the terminal aryl ring is involved in π–π stacking interactions with Tyr221. In the S configuration of **6d**, there is a clash between the ethyl substituent and the oxygen of Ser64 (2.9 Å), as well as close contact with Leu119 (3.3 Å). In **6e**, there is also a clash between the ethyl group (3.1 Å) and Ser64; however, here there is a favorable van der Waals interaction with Leu119, as well as a favorable stacking interaction between Tyr221 and the aryl ring of **6e**. 

For the complexes with OXA-24/40, all three inhibitors are positioned such that the bridge formed by the Tyr112 and Met223 side chains is intact (Figure 7). In its complex with OXA-24/40, **6d** (the *S* configuration) binds in a similar conformation as **CR167**, resembling that of doripenem in its complex with OXA-24/40 (Figure 3A,B). The carboxylate of **6d** interacts with Arg261 (distances: 2.7–2.9 Å) and also forms a hydrogen bond with Ser219. The sulfonamide nitrogen forms a hydrogen bond with the main chain carbonyl oxygen of Trp221 (3.1 Å). The edge-to-face interaction between Tyr112 and the aryl ring of **6d** is maintained, remaining essentially the same as that observed in the **CR167** complex, with a centroid–centroid distance of 5.5 Å and an angle between the aromatic rings of 89°. The ethyl substituent of **6d** is positioned in a hydrophobic pocket, and it formed van der Waals interactions with the side chain of Val130 (3.4 Å) in one of two observed conformers for this residue. However, the ethyl group is in close contact with the carbamylated lysine (KCX84; 3.3 Å) and the main chain carbonyl oxygen of Trp221 (3.1 Å), as well as the side chain of Leu168 (3.0 Å). In the complex of OXA-24/40 with **6e** (Figure 7), the *R* configuration of the stereogenic carbon atom determines several differences. The most significant is that **6e** adopts a trajectory that places the inhibitor in the R1 binding site of the active site versus the R2 site observed with **CR167** and **6d**. In this orientation, the carboxylate group of **6e** is not positioned to interact with Arg261. Instead, the carboxylate forms hydrogen bonds with the side chain hydroxyl of Thr111 and a water molecule (426). The sulfonamide nitrogen again forms a hydrogen bond with the main chain carbonyl oxygen of Trp221 (3.0 Å). In this conformation, the ethyl substituent of **6e** is oriented approximately 180° from its location in the **6d** complex, where it maintains an interaction with Val130 (3.3 Å), in one of its conformers, but it is in close contact with the main chain carbonyl oxygen of Ser128 (3.3 Å).

Interestingly, in both OXA-24/40 complexes with the chiral compounds, Val130 adopts two distinct alternate conformations, which is not the case in the OXA-24/40/**CR167** complex. Rotation of this residue is likely driven by close contacts with the ethyl substituent of the chiral inhibitors, with the clash occurring in only one of the two Val130 conformations, which explains the better binding affinity for inhibitors containing the smaller methyl substituent.

Antimicrobial susceptibility testing

Based upon the inhibitor kinetics (*K*i determinations), **CR167** and the sulfonamidoboronic acid **6b** were tested against *E. coli* strains expressing ADC-7 and ADC-33 cloned into pBCSK- (DH10B) (Table 3). In this set of experiments, both compounds lowered the CAZ MICs by four doubling dilutions: for ADC-7, MIC = 64 mg/L CAZ to 4 mg/L CAZ/**CR167** and CAZ/**6b**; for ADC-33, MIC = 2048 mg/L CAZ to 64 mg/L CAZ/**CR167** and CAZ/**6b**. Cefotaxime (CTX) MICs were lowered by five to six doubling dilutions: for ADC-7, MIC = 64 mg/L CTX to 2 mg/L CTX/**CR167** and CTX/**6b**; for ADC-33, MIC = 128 mg/L CTX to 2 mg/L CTX/**CR167** and 4 mg/L CTX/**6b**. Cefepime (FEP) MICs were also lowered when coupled with both inhibitors; however, both isolates were very susceptible to FEP initially. AST determinations were not undertaken for OXA-24/-40 in *E. coli* as we did not have a construct that expressed well against that background.

The inhibitory effects of both compounds were tested against ADC-33 and OXA-24/-40 in *A. baumannii*, their native host. ATCC 17978, with its intrinsic chromosomal ADC knocked out (intrinsic OXA still present), was used in these experiments. In this isolate, the ADC MICs were not lowered by the inhibitors as much as they were against the *E. coli* background (Table 4). However, a decrease in the MIC was observed for ADC-33, with the MIC of CAZ declining from 128 mg/L to 32 mg/L with the addition of **CR167** and the MIC of CTX declining from 32 mg/L to 16 mg/L with the addition of **CR167**. No observable diminutions in the MICs were observed for the OXA-24/-40 containing isolate. 

Clinical *A. baumannii* isolates containing ADC-33 or OXA-24/-40 in combination with various other well-defined *β*-lactamases were also tested against **CR167** and **6b** (Table 5). In five out of seven isolates, a one- to twofold dilution decrease was observed for CAZ in combination with **CR167**; in four out of seven isolates, a one- to twofold dilution decrease was observed for CTX in combination with **CR167**. Regarding the BATSI **6b**, two out of seven isolates showed a onefold dilution decrease in MICs for CAZ or CTX.

## 3. Discussion

*A. baumannii* is one of the most difficult pathogens to treat in clinical practice, especially because it is often resistant to all classes of antibiotics. The evidence showing that treatment options against *A. baumannii* infections are limited makes *β*-lactamase inhibition an important strategy to preserve the efficacy of currently available antibiotics. To face often pan-drug-resistant organisms, such as *A. baumannii*, it is crucial to consider the most common *β*-lactamases occurring in the pathogen rather than focusing on a single class of enzymes and to identify a cross-class inhibitor with activity against multiple *β*-lactamase classes. To reach this objective, we explored the activity of the inhibitor **CR167** against critical ADC (class C) and OXA (class D) enzymes. 

Indeed, **CR167** is a boronic acid transition state inhibitor (BATSI) that was initially synthesized as an inhibitor of class A and class C *β*-lactamases. **CR167** demonstrated potent activity against several critical enzymes of class C (*K*_i_ for AmpC = 0.0012 μM) and class A (IC_50_ for SHV-1 = 0.43 μM and IC_50_ for KPC-2 = 0.20 μM). Additionally, **CR167** was shown to inhibit ADC-7 with nM activity (*K*_i_ = 160 nM).

To explore the activity of such an interesting compound for cross-class inhibition more deeply, we tested it against two representative β-lactamases, the class C ADC-33 and the class D OXA-24/40, that are prevalent in *A. baumannii*. The binding mode of **CR167** was then compared with that of doripenem and, using the complex of OXA-24/40 with **CR167** as our template, five chiral benzylsulfonamidoboronic acid analogues of **CR167** were synthesized. We introduced a small hydrophobic substituent (methyl or ethyl) on the boron-bearing carbon atom, which was meant to mimic the hydroxyethyl side chain of doripenem, and we synthesized compounds with and without an *m*-carboxylate on the benzylsulfonyl group to assess the importance of a negative charge for recognition in class C and D *β*-lactamases. 

Based on the results of these experiments, the best-performing compound/combinations were **CR167** (*K*_i_ for ADC-33 = 0.057 μM, *K*_i_ for OXA-24/40 = 4.4 μM; MIC = 128 mg/L CTX to 2 mg/L CTX/**CR167** for *E. coli* DH10B/ADC-33 and MIC = 64 mg/L CTX to 2 mg/L CTX/**CR167** for *E. coli* DH10B/ADC-7) and compound **6b** (*K*_i_ for ADC-33 = 0.79 μM, *K*_i_ for OXA-24 = 5.2 μM; MIC = 128 mg/L CTX to 4 mg/L CTX/**6b** for *E. coli* DH10B/ADC-33 and MIC = 64 mg/L CTX to 2 mg/L CTX/**6b** for *E. coli* DH10B/ADC-7), which differed only in the methyl group at the carbon next to the boron atom. Substituents at this position were meant to mimic the hydroxyethyl group found in carbapenems. This suggested that a methyl side chain inserted in the **6b** structure could be accommodated in the small OXA-24/40 site without significant loss (or gain) of activity. Significant lowering of the CTX MICs of the *bla*_ADC-33_ and *bla*_OXA-24/-40_ constructs in *A. baumannii* ATCC17978 ADC KO was not observed. However, some efficacy was noted with the combination of CAZ and **CR167**. The CAZ MICs decreased from 128 mg/L to 32 mg/L with the addition of 10 mg/L **CR167**. Clinical isolates also did not show a large reduction in MICs with either compound. However, in most clinical isolates, a one- to twofold dilution decrease was observed for CAZ and CTX in combination with **CR167**. Penetration of these compounds into the cell and their efflux out of it are being actively investigated.

The X-ray crystal structures of compounds **6d** and **6e** bearing the ethyl side chain corroborated the results obtained in other assays. For ADC-33, a class C β-lactamase, the (*R*) configuration in **6e** of the chiral center alpha to the boron atom is preferred over the (*S*) configuration of **6d**, as the ethyl chain in (*S*)-**6d** clashes with Ser64 and Leu119. The smaller methyl substituent would not result in these clashes with the enzyme. However, inhibitors **CR167**, **6d,** and **6e** were, overall, well-accommodated in the ADC binding pocket, confirming the orientation previously described for the ADC-7/**CR167** complex. The sulfonamide behaves as a bioisostere of the C7-amide chain of cephalosporins and interacts with key residues Asn152 and Gln120, while the benzyl ring stacks with Tyr221. The structures of all three compounds extend in the R1 region of the enzyme and no additional interactions are displayed by the hydrophobic chain introduced on the carbon α to the boron. 

In contrast, for OXA-24/40, the two chiral compounds are oriented in completely different binding modes in the enzymatic pocket. The *S* configuration of **6d** is preferred over the *R* in **6e**, as **6d** maintains the interaction with Arg261 that is fundamental in doripenem binding and also present in the **CR167** complex. In **6e**, the carboxylate group clashes with Val130 in its “open” conformation and is in close contact with Thr111. Due to the tetrahedral geometry around the boron atom of the chiral and achiral BATSIs, the carbon α to the boron is angled such that it is ~1 Å away from the C6 carbon that bears the hydroxyethyl side chain of doripenem. The consequence of this shift places the ethyl substituent of **6d** deeper in the hydrophobic pocket formed by Val130, Trp167, Leu168, and Val169 and, surprisingly, in close contact with the side chain of Leu168, even clashing with the main chain carbonyl oxygen of Trp221. The ethyl substituent in **6e** is oriented away from this pocket, and although it does not clash with the enzyme, it is not stabilized either. This again highlights the importance of maintaining an interaction with Arg261 in the OXA enzymes.

## 4. Materials and Methods

### 4.1. Chemistry

All reagents and solvents were purchased from Sigma-Aldrich, Fluorochem, and Enamine. Anhydrous tetrahydrofuran (THF) was obtained with the standard method and freshly distilled under Ar from sodium benzophenone ketyl prior to use. The −100 °C bath was prepared through the addition of liquid nitrogen to a precooled (−80 °C) mixture of ethanol/methanol (1:1). Reactions were monitored with TLC and visualized using UV fluorescence and Hanessian’s cerium molybdate stain. Chromatographic purification of the compounds was performed on silica gel (particle size: 0.05–0.20 mm). Optical rotations were recorded at +20 °C on a Perkin-Elmer 241 polarimeter and are expressed in 10^−1^deg cm^2^ g^−1^. ^1^H and ^13^C NMR spectra were recorded on a Bruker Avance-400 spectrometer. Chemical shifts (*δ*) are reported in ppm downfield from tetramethylsilane (TMS) as the internal standard (s, singlet; d, doublet; t, triplet; q, quartet; m, multiplet). Coupling constants (*J*) are given in Hz. Two-dimensional NMR techniques (HMBC, HSQC) were used to help in the assignment of signals in ^1^H and ^13^C spectra. High-resolution mass spectra were recorded on an LC-MS apparatus: a Thermo Scientific UHPLC Ultimate 3000 coupled with a Q Exactive™ Hybrid Quadrupole-Orbitrap™ Mass Spectrometer. 

### 4.2. Expression, Purification, and Crystallization of ADC-33 and OXA-24/40

ADC-33 was expressed and purified using the same methods as those reported for ADC-7 [37]. Crystallization of ADC-33 was performed using hanging drop vapor diffusion at room temperature with ADC-33 (3 mg/mL) in a well buffer containing 0.1 M succinate/phosphate/glycine, pH 5.0, 25% PEG-1500. OXA-24/40 was expressed, purified, and crystallized as previously reported [33,34].

### 4.3. Structure Determination of β-Lactamase/BATSI Complexes

For ADC-33/BATSI complexes, the boronic acids were prepared at a concentration of 5 mM in well buffer (0.1 M succinate/phosphate/glycine, pH 5.0, 25% *w*/*v* PEG-1500). Aliquots of these solutions were added directly to crystallization drops containing pre-formed ADC-33 crystals and allowed to equilibrate for 30–45 min. For OXA-24/40 complexes, crystals were harvested from drops and soaked in a 5–10% sucrose cryoprotectant solution of well buffer (100 mM HEPES pH 7.5, 2% *v*/*v* PEG-400, 2.0 M ammonium sulfate) supplemented with 25 mM bicarbonate and either **CR167** (1 mM) or **6d/6e** (5 mM) for 30–45 min. Crystals were then flash cooled in liquid nitrogen, stored in pucks, and shipped to the Advanced Photon Source at Argonne National Lab. Data were measured from single crystals at LS-CAT (beamline 21ID-F for **6d** and **6e** or beamline 21ID-D for **CR167**). Diffraction data were processed with XIA2 [38], except for the **CR167** complex, which was processed with Autoproc [39]. Structures were determined by molecular replacement with the program Phaser [40], using as a starting model either an apo structure of ADC-33 (not published) or a K84D variant of OXA-24/40 (3PAE) with all ligands, ions, and waters removed. Models were refined with Phenix [40], followed by rounds of model building in Coot [41]. Polder omit electron-density maps were also calculated with Phenix. Coordinates and structure factors were deposited in the Protein Data Bank (PDB) with the following accession codes: OXA-24/40 complexes with **CR167** (8CUL), **6d** (8CUM), and **6e** (8CUO); ADC-33 complexes with **6d** (8CUP) and **6e** (8CUQ).

### 4.4. Determination of K_i_ Values

*K*_i_ values were determined by using nitrocefin as a reporter substrate and treating the compounds as competitive inhibitors. Steady-state kinetic parameters were determined by combining pure enzyme with nitrocefin in 50 mM NaH_2_PO_4_; 25 mM NaHCO_3_, pH 7.4 (for OXA-24/40); or 50 mM NaH_2_PO_4_, pH 7.4 (for ADC-33), at room temperature. Changes in absorbance were measured on a Cary 60 UV−Vis spectrophotometer (Agilent Technologies) and converted to velocity using the change-in-extinction coefficient specific to nitrocefin (ε = 17,400 M^−1^·cm^−1^ at 482 nm). Initial velocities were fit to the Michaelis–Menten equation, yielding *k*_cat_ and *K*_m_ values. Next, the enzyme was challenged with a constant concentration of nitrocefin and increasing amounts of inhibitor. IC_50_ values were used to calculate *K*_i_ values using the Cheng–Prusoff equation as previously described [42]. Specifically, the measurements of the initial velocities were performed with the addition of 100 µM NCF after a 5 min pre-incubation of the enzyme (2 nM) with increasing concentration of the inhibitor [37]. To determine the average velocities (*v_0_*), data from three experiments were fit to Equation (1): (1)$$v_{0}=v_{u\;}-\left\{\frac{v_{u}\left[I\right]}{IC_{\;50}+\left[I\right]}\right\}$$
where *v_u_* represents the NCF uninhibited velocity and *IC*_50_ represents the inhibitor concentration that results in a 50% reduction in *v_u_*. The *K_i_* value was corrected for the NCF affinity (*K*_m_ values for both OXA-24/40 and ADC-33) with the Cheng–Prusoff equation (Equation (2)): (2)$$K_{i}=IC_{50}/\left(1+\frac{\left[NCF\right]}{K_{mNCF}}\right)$$

### 4.5. Plasmid Constructs in pBCSK-

*bla*_ADC-7_ pBCSK- and *bla*_ADC-33_ pBCSK- were cloned and expressed in *E. coli* DH10B cells as previously described [11]. 

### 4.6. Cloning of bla_ADC-33_ for Expression in A. baumannii

Plasmid pBCSK- containing the *bla*_ADC-33_ gene was digested with PvuII, which released the *bla*_ADC-33_ gene, along with the *lac* promoter, which drives expression. This PvuII fragment was cloned into the SmaI site of the pQF1266 plasmid [43].

### 4.7. Cloning the bla_OXA-24/-40_ for Expression in A. baumannii

pWH1266 is a vector that replicates in *A. baumannii*, and the *bla*_ADC-24/-40_ pWH1266 construct was generated as follows. The TEM gene was removed using inverse PCR, and an Xba1 site was engineered into the vector. *bla*_OXA-24/40_ and its upstream XerC/XerD were amplified from strain NM55. The amplified gene was cloned into the Xba1 site of the altered pWH1266 vector. The construct was confirmed using PCR and sequencing. Both *bla*_ADC-33_ pQF1266 and *bla*_ADC-24/-40_ pWH1266 were then expressed in *A. baumannii* ATCC 17978 that had the intrinsic chromosomal ADC gene knocked out.

### 4.8. Construction of the bla_ADC-51_ Deletion Mutant of ATCC 17978 

The chromosomal *bla*_ADC-51_ gene in ATCC 17978 was deleted and replaced with an apramycin resistance cassette using the recombineering method described by Tucker et al. [44]. Primers used for this deletion were: Del AmpC-apra.For: TCGACCAATTCTATAAAATATAACAATTAAATTAGGTGATTTTTGTTATAAAAGTAGGCATCTTTCTTTTTAAATAATTTATGAGCTAATCATGCGATTTATGTCATCAGCGGTGGAGTGCAATG; and Del AmpC-apra.Rev: GCTTAGGATATGTTTGGTTCTTTTAAAGAATAAGTGCATAAAAAAGAGAAATGGGATTATCCATTTCTCTTTTTGTTTTTGGCTAAAACGGTTATTTCTTTCAGCCAATCGACTGGCGAGCGGC. The correct deletion mutant was confirmed using PCR and sequencing.

### 4.9. Selection of Clinical Bacterial Isolates

The clinical *A. baumannii* isolates used to test the β-lactamase inhibitors were selected from a previously well-characterized collection in our lab [8,45]. They were chosen as they had either ADC-33 and/or OXA-24 in combination with other known β-lactamases (for specific combinations, see Table 5).

### 4.10. Antimicrobial Susceptibility Testing (AST)

Testing of susceptibility to standard antibiotics was performed with broth microdilution according to CLSI guidelines. MICs for CAZ, CTX, FEP, and IMI were determined using cation-adjusted Mueller–Hinton broth according to CLSI methods, and **CR167** or **6b** was used at a fixed concentration of 10 mg/L. All MICs were interpreted according to the 2022 Clinical and Laboratory Standards Institute (CLSI) guidelines [45]. 

## 5. Conclusions

Analysis of the BATSI **CR167** and its chiral analogs **6a–e** allows us to highlight some properties necessary for cross-class C and D inhibition. The sulfonamide group is a good substitute of the more explored carboxamide for the inhibition of both enzyme classes. In ADC, it forms hydrogen bonds with key residues Asn152 and Gln120, while in OXA-24/40, it allows the binding of the boronic acid without clashing with the hydrophobic bridge formed by Tyr112 and Met223 and allows for a trajectory such that the carboxylate group can form a favorable ionic interaction with Arg261. In this way, the compounds can exert their inhibitory activity. 

In chiral derivatives, the stereochemical preferences of the two enzymes are opposite. In ADC, the *R* enantiomer is better accommodated than the *S* (**6e** vs **6d**), mimicking the configuration at C7 of cephalosporins. In contrast, in OXA-24/40, the preferred configuration is *S* (**6d** is better than **6e**), reproducing the stereochemistry at C6 of the carbapenem doripenem, although the boronic acid tetrahedral geometry shifts the carbon atom alpha to the boron closer to the carboxylated lysine KCX84 (by ~1 Å), resulting in a more constricted region for the added substituent (i.e., the methyl substituent was preferred to the ethyl). The carboxylate group on the benzylsulfonamide provides a greater contribution to binding affinity in OXA-24/40 than in ADC. In ADC, it interacts with Asn212 and Ser320 on the opposite side with respect to the site where the C3 (C4) carboxylate of *β*-lactams is placed. In OXA-24/40, the carboxylate group of **CR167** forms favorable ionic interactions with the carboxylate recognition residue Arg261. The absence of this anionic group (as in compounds **6a** and **6c**) reduced the activity against both *β*-lactamases: against OXA-24/40 **6a,** it was 121-fold less potent than **6b,** whilst it was only 4-fold less potent against ADC-33. Finally, the smaller the group attached to the boron-bearing carbon atom is (hydrogen in **CR167**, methyl in **6b**, ethyl in **6d**), the better the binding. Bearing in mind that this group was introduced to mimic the hydroxyethyl side chain in C6 of carbapenems and that the corresponding binding region in OXA enzymes is very restricted in size, the methyl group alleviates the clashes observed for the more sterically demanding ethyl group.

This work provides some guidelines and a lead scaffold to help in the future development of new sulfonamide boronic acids with activity against not only class C and class A *β*-lactamases but also against critical class D enzymes. To this end, the introduction of hydrophobic substituents on the benzylsulfonyl group will be explored to increase the compound’s stability and reduce its susceptibility to phase 1 metabolism. Further studies are also needed to improve the ability of these compounds to permeate the outer membrane of *A. baumannii* and overcome its resistance mechanisms.

## Data Availability

Not applicable.

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
