# Peer review of "Sulfonamidoboronic Acids as “Cross-Class” Inhibitors of an Expanded-Spectrum Class C Cephalosporinase, ADC-33, and a Class D Carbapenemase, OXA-24/40: Strategic Compound Design to Combat Resistance in Acinetobacter baumannii"

_antibiotics, 2023, doi:10.3390/antibiotics12040644_

Round 1

Reviewer 1 Report

Introvigne et al. described that CR167 is a inhibitor against the β-lactamases ADC-33 and OXA-24/40. They designed 5 chiral analogues of CR167. The structures of ADC-33 and OXA-24/40 in complex with CR167 and CR167 analogues were obtained. The determinants for cross-class C/D inhibitors were suggested. The manuscript is currently in its good shape.

Author Response

No revision required: thank you!

Reviewer 2 Report

The manuscript presents an intricate analysis of the application of Sulfonamidoboronic acid as inhibitors of Acinetobacter-derived Cephalosporinase-33 (ADC-33) and oxacillin-hydrolyzing enzymes (OXA-24/40), which are categorized under class C and D β-lactamases exhibiting resistance to β-lactam antibiotics. As a follow-up to the authors' previous research, the authors delved into the sulfonamidomethaneboronic acid CR167's activity, along with its chiral analogues, against ADC-33 and OXA-24/40, rendering the likelihood of cross-class treatment for Acinetobacter baumannii infection. This manuscript shall appeal to researchers and clinicians investigating the development of novel therapies against antibiotic-resistant bacterial infections. I have some minor objections as stated below.

1. Figure 4/5/6/7: repetitive image.

2. Please include the PDB validation reports concerning the structures.

Author Response

"The manuscript presents an intricate analysis of the application of Sulfonamidoboronic acid as inhibitors of Acinetobacter-derived Cephalosporinase-33 (ADC-33) and oxacillin-hydrolyzing enzymes (OXA-24/40), which are categorized under class C and D β-lactamases exhibiting resistance to β-lactam antibiotics. As a follow-up to the authors' previous research, the authors delved into the sulfonamidomethaneboronic acid CR167's activity, along with its chiral analogues, against ADC-33 and OXA-24/40, rendering the likelihood of cross-class treatment for Acinetobacter baumannii infection. This manuscript shall appeal to researchers and clinicians investigating the development of novel therapies against antibiotic-resistant bacterial infections. I have some minor objections as stated below."

  1. Figure 4/5/6/7: repetitive image.

Each of these images conveys a specific point in the narrative of the manuscript and speaks to the evolution of our design optimization efforts of cross-class inhibitors. An attempt was made to combine Figures 4 and 5 into a single image, but this severely impacted the clarity of the image. Respectfully, we ask to keep the images as submitted.

  1. Please include the PDB validation reports concerning the structures.

The reports are ready will be uploaded as required.Sorry for this forgetfulness

Reviewer 3 Report

1.     On line 432, It would be nice if authors stated the significance of the introduction of small hydrophobic substituents and carboxylates on the benzylsulfonyl group in the synthesized compounds.

2.     In section Inhibition Kinetics i.e., line 287, Please explain the rationale for determining the incubation time? why <5 mins ? How authors arrived at this time period for incubation?

3.     Please comment on the following :
How do authors anticipate the modifications of Sulfonamidoboronic during Phase 1 metabolism in the liver? This is a crucial parameter as it can significantly alter the chemical nature of the compound, potentially affecting its binding affinity and effectiveness against beta-lactamases. Therefore, careful consideration must be given to this aspect of drug development. As such, the authors may be exploring the introduction of hydrophobic substituents on the benzylsulfonyl group to increase the compound's stability and reduce its susceptibility to Phase 1 metabolism. This approach may help to enhance the drug's efficacy and prolong its half-life in the body, leading to improved clinical outcomes.

4.     In certain figures depicting the interactions between CR167 and OXA-24/40, the interactions may not be immediately clear to the reader. To enhance the clarity of these interactions, it is suggested that the authors consider using a darker color to represent these interactions. This approach would aid in highlighting the key interactions and enable the reader to easily comprehend the information being conveyed. By doing so, the reader would be able to better appreciate the significance of these interactions in the context of the study's objectives.

Author Response

  1. On line 432, It would be nice if authors stated the significance of the introduction of small hydrophobic substituents and carboxylates on the benzylsulfonyl group in the synthesized compounds.

Thank you for this suggestion! We modified lines 434-437 as follows:

We introduced a small hydrophobic substituent (methyl or ethyl) on the boron-bearing carbon atom, which is meant to mimic the hydroxyethyl side chain of doripenemand we synthesized compounds with and without a m-carboxylate on the benzylsulfonyl group to assess the importance of a negative charge for recognition in class C and D β-lactamases. 

  1. In section Inhibition Kinetics i.e., line 287, Please explain the rationale for determining the incubation time? why <5 mins ? How authors arrived at this time period for incubation?

Thank you for pointing this out. The incubation time was indeed kept at 5 minutes as usually reported (for example see: Caselli et al, J Med Chem, 2015, 58(14), 5445-5458). We made the proper corrections in the text

  1. Please comment on the following:

How do authors anticipate the modifications of Sulfonamidoboronic during Phase 1 metabolism in the liver? This is a crucial parameter as it can significantly alter the chemical nature of the compound, potentially affecting its binding affinity and effectiveness against beta-lactamases. Therefore, careful consideration must be given to this aspect of drug development. As such, the authors may be exploring the introduction of hydrophobic substituents on the benzylsulfonyl group to increase the compound's stability and reduce its susceptibility to Phase 1 metabolism. This approach may help to enhance the drug's efficacy and prolong its half-life in the body, leading to improved clinical outcomes. We thank the reviewer again for the thoughtful suggestions. We have added this consideration in the text.

       -line 614:

This work provides some guidelines and a lead scaffold to help in the future development of new sulfonamide boronic acids with activity against not only class C and class A β-lactamases but also against critical class D enzymes. To this end the introduction of hydrophobic substituents on the benzylsulfonyl group will be explored to increase the compound's stability and reduce its susceptibility to Phase 1 metabolism.

  1. In certain figures depicting the interactions between CR167 and OXA-24/40, the interactions may not be immediately clear to the reader. To enhance the clarity of these interactions, it is suggested that the authors consider using a darker color to represent these interactions. This approach would aid in highlighting the key interactions and enable the reader to easily comprehend the information being conveyed. By doing so, the reader would be able to better appreciate the significance of these interactions in the context of the study's objectives.

We changed the color of the dashed lines that indicate interactions between the enzyme and the inhibitor in Figures 4 and 7 to dark gray (from the original light yellow).

Reviewer 4 Report

The manuscript "Sulfonamidoboronic acids as “cross-class” inhibitors of an Expanded-spectrum Class C cephalosporinase, ADC-33, and a Class D carbapenemase, OXA-24/40: strategic compound design to combat resistance in Acinetobacter baumannii" presented by Introvigne ML et al. is well-written with clear logic. The results are solid and clear with few minor issues. 

Comments/suggestions:

1.     The colors in Figure 3 & Figure 7 are not distinguishable, please modify.

2.     In Figure 4, please highlight ligand with different color.

3.     In Figure 7, please highlight the ligand as well as the interaction between the ligand and Arg261.

4.     Line 577, please add reference. 

Author Response

  1. The colors in Figure 3 & Figure 7 are not distinguishable, please modify.

We updated both Figures 3 and 7 by changing the carbon atom coloring scheme to one that is more distinct for the reader. Carbon atoms of the OXA-24/40/CR167 complex are now a darker smudge green, and the carbons for the OXA-24/40/doripenem complex are now slate blue.

  1. In Figure 4, please highlight ligand with different color.

Figure 4 has also been updated to highlighting the inhibitor bound in the active site by coloring the carbon atoms of the inhibitor a different shade of green (forest), which differs from the carbon atoms of the enzyme residues (smudge).

  1. In Figure 7, please highlight the ligand as well as the interaction between the ligand and Arg261.

To highlight the ligand, interactions between the inhibitor and Arg261 have been added and are indicated with dashed gray lines.

  1. Line 577, please add reference. The reminder “(add reference for CLSI)” in line 577 of the original paper was a typo.

Thank you for noticing this…the reference is number 48, already present at the end of the sentence, and I simply forgot to cancel my reminder...sorry!